# Analysis of *CDO1*, *PITX2*, and *CDH13* Gene Methylation in Early Endometrial Cancer for Prediction of Medical Treatment Outcomes

**DOI:** 10.3390/ijms25094892

**Published:** 2024-04-30

**Authors:** Aleksey M. Krasnyi, Lyubov T. Gadzhieva, Diana N. Kokoeva, Mark G. Kosenko, Ekaterina L. Yarotskaya, Stanislav V. Pavlovich, Levon A. Ashrafyan, Gennady T. Sukhikh

**Affiliations:** 1V.I. Kulakov National Medical Research Center for Obstetrics, Gynecology and Perinatology, 117198 Moscow, Russia; 2Moscow Multidisciplinary Clinical Center “Kommunarka”, 108814 Moscow, Russia; 3I.M. Sechenov First Moscow State Medical University, 119991 Moscow, Russia

**Keywords:** methylation, atypical endometrial hyperplasia, endometrial cancer stage IA, organ-preserving treatment, MS-HRM, *CDO1*, *CDH13*

## Abstract

An observational cohort study of patients diagnosed with endometrial cancer (EC) stage IA G1, or atypical endometrial hyperplasia (AEH), undergoing organ-preserving treatment, was conducted. Objective of the study: To determine *CDO1*, *PITX2*, and *CDH13* gene methylation levels in early endometrial cancer and atypical hyperplasia specimens obtained before organ-preserving treatment in the patients with adequate response and with insufficient response to hormonal treatment. Materials and methods: A total of 41 endometrial specimens obtained during diagnostic uterine curettage in women with EC (*n* = 28) and AEH (*n* = 13), willing to preserve reproductive function, were studied; 18 specimens of uterine cancer IA stage G1 from peri- and early postmenopausal women (comparison group) were included in the study. The control group included 18 endometrial specimens from healthy women obtained by diagnostic curettage for missed abortion and/or intrauterine adhesions. Methylation levels were analyzed using the modified MS-HRM method. Results: All 13 women with AEH had a complete response (CR) to medical treatment. In the group undergoing organ-preserving treatment for uterine cancer IA stage G1 (*n* = 28), 14 patients had a complete response (EC CR group) and 14 did not (EC non-CR group). It was found that all groups had statistically significant differences in *CDO1* gene methylation levels compared to the control group (*p* < 0.001) except for the EC CR group (*p* = 0.21). The *p*-value for the difference between EC CR and EC non-CR groups was <0.001. The differences in *PITX2* gene methylation levels between the control and study groups were also significantly different (*p* < 0.001), except for the AEH group (*p* = 0.21). For the difference between EC CR and EC non-CR groups, the *p*-value was 0.43. For *CDH13* gene methylation levels, statistically significant differences were found between the control and EC non-CR groups (*p* < 0.001), and the control and EC comparison groups (*p* = 0.005). When comparing the EC CR group with EC non-CR group, the *p*-value for this gene was <0.001. The simultaneous assessment of *CDO1* and *CDH13* genes methylation allowed for an accurate distinction between EC CR and EC non-CR groups (AUC = 0.96). Conclusion: The assessment of *CDO1* and *CDH13* gene methylation in endometrial specimens from patients with endometrial cancer (IA stage G1), scheduled for medical treatment, can predict the treatment outcome.

## 1. Introduction

Endometrial cancer (EC) in young women of reproductive age is rare: only 4% of patients are under 40 years of age [1]. The standard treatment for endometrial cancer is hysterectomy and bilateral salpingo-oophorectomy with or without lymphadenectomy. However, for patients who wish to preserve their reproductive function, conservative treatment for EC, based on progestins, is possible. FIGO guidelines suggest that conservative treatment can be considered in patients with a histologically confirmed well-differentiated endometrioid adenocarcinoma G1, or atypical hyperplasia (AEH) [2]. Patients undergoing conservative treatment may experience unfavorable outcomes such as an absence of treatment effect, disease recurrence, and disease progression [3]. The expression of progesterone receptors has been considered as one of the predictors of complete response, but in a study by Yamazawa et al., only 50% effectiveness with a specificity of 100% was observed for this treatment option [4]. An analysis of the expression of mismatch repair (MMR) proteins showed even lower effectiveness (22%) [5]. Today, there are no reliable prognostic markers for use in clinical practice [6]. Methylation changes of some genes are associated with a poor prognosis related to cancer treatment. The study by Hirano et al. on genome-wide DNA methylation profiles, showed that more aggressive endometrial tumors in young women had a specific methylation profile [7]. One of the key factors in cancer development is the mutations or epigenetic disorders of tumor suppressor genes (TSG). These genes control cell development in the growth and division cycles and maintain genome integrity, while their silencing promotes cancer cell survival and proliferation [8]. These facts allowed us to suggest that epigenetic silencing of TSG may influence the outcomes of EC treatment. The aim of this study was to examine the methylation of certain TSGs as potential markers for predicting EC treatment outcomes.

## 2. Results

All 13 women with atypical endometrial hyperplasia had a complete response to the treatment, and the disease did not progress (AEH group). Among the women with uterine cancer, 14 patients had a complete response (EC CR group) and 14 patients had a non-complete response (EC non-CR group): partial response (*n* = 3); stable disease (*n* = 2); disease recurrence within one year after a successful 6-month course of treatment (*n* = 5); or disease progression (*n* = 4). The age and BMI of the patients are presented in Table 1. In our study, we did not find significant differences in BMI between the EC CR group and the EC non-CR group. The number of obese patients (BMI > 30) in the EC CR group was ten (71%), and in the EC non-CR group was eight (57%) (*p* = 1).

Patients from the EC non-CR group with either stable disease or disease progression underwent hysterectomy; the rest underwent repeated courses of therapy. Four of them had atrophy of all endometrial glands after two additional treatment courses, and subsequently underwent in vitro fertilization and embryo transfer (IVF&ET), as did nine and seven patients from the AEH and EC non-CR groups, respectively. The outcomes are presented in Table 2 and demonstrate that in the case of complete response within 6 months for both AEH and EC, about 50% of patients, who had attempted to have children, gave birth to live children. However, if the complete response had been achieved only after additional courses of treatment, IVF&ET was not successful.

To study the effect of TSG methylation on the outcomes of the conservative treatment of patients with endometrioid adenocarcinoma, eight genes generally acknowledged as TSGs were initially selected: *P16*, *APC*, *CDH13*, *SEPTIN9*, *PITX2*, *SHOX2*, *CDO1*, and *PTEN*.

However, the comparison of methylation levels of these genes between the control and comparison groups confirmed significant differences only for *CDH13*, *PITX2*, and *CDO1* genes (Figure 1, Figure 2 and Figure 3). Therefore, only three genes were included in our study, while the methylation data for genes that did not show differences are not presented.

The methylation of the *CDO1*, *PITX2*, and *CDH13* genes was carried out in the AEH and EC groups, in the control group (healthy women), and the comparison group (women with uterine cancer stage IA in premenopause and early postmenopause).

It was found that, in all groups, *CDO1* methylation levels were significantly different from the control group (*p* < 0.001), except for the EC CR group (*p* = 0.21). The *p*-value for the difference between EC CR and EC non-CR groups was <0.001. Methylation levels were as follows: control group—0.001 (0.001; 0.003) rel. units; AEH group—0.021 (0.021; 0.041) rel. units; EC CR group—0.012 (0.001; 0.101) rel. units; EC non-CR group—0.241 (0.171; 0.346); and EC, comparison—0.146 (0.091; 0.263) rel. units (Figure 1).

The assessment of *PITX2* methylation level showed that all groups had statistically significant differences compared to the control group (*p* < 0.001) except for the AEH group (*p* = 0.21). For the difference between EC CR and EC non-CR groups, the *p*-value was 0.43. Methylation levels were as follows: control group—0.025 (0.018; 0.035) rel. units; AEH group—0.03 (0.02; 0.075) rel. units; EC CR group—0.045 (0.037; 0.07) rel. units; EC non-CR group—0.085 (0.035; 0.21); and EC, comparison—0.056 (0.035; 0.245) rel. units (Figure 2).

*CDH13* gene methylation levels were significantly different between the control and EC non-CR groups (*p* < 0.001), and the control and EC comparison groups (*p* = 0.005). When comparing the EC CR group with the EC non-CR group, the *p*-value for this gene was <0.001. Methylation levels were as follows: control group—0.03 (0.02; 0.03) rel. units; AEH group—0.03 (0.02; 0.04) rel. units; EC CR group—0.03 (0.02; 0.04) rel. units; EC non-CR group—0.16 (0.07; 0.187); and EC, comparison—0.09 (0.03; 0.15) rel. units (Figure 4).

ROC analysis showed a high diagnostic value for the *CDH13* gene, with AUC = 0.88 (0.75–1), and for the *CDO1* gene, with AUC = 0.9 (0.89–1). Logistic regression for both genes showed AUC = 0.96 (0.89–1). ROC curves are presented in Figure 4.

## 3. Discussion

In the treatment of early EC, attempting to preserve fertility has become possible, since this disease is most likely to have a favorable prognosis. However, some patients experience a recurrence or progression of the disease. In the latter case, urgent radical surgery is necessary due to the increased risk to the patient’s life. Several prognostic molecular markers for treatment outcomes have been proposed. The most studied are progesterone receptors; however, the related findings are contradictory. Ki67, Nrf2, SPAG9, and MMR proteins were also considered as prognostic markers, but they did not show high diagnostic value [6].

Some researchers believe that excess body weight (obesity) is a risk factor for worse outcomes of medical treatment [9,10]. However, in our study, this factor had no significant differences between the EC CR group and the EC non-CR group.

The assessment of reproductive outcomes in patients with additional treatment courses showed that all patients experienced missed abortion. This finding requires further research, since it raises questions about the advisability of additional treatment courses for EC.

In this study, we determined *CDO1*, *PITX2*, and *CDH13* gene methylation levels in the EC and AEH specimens of patients before conservative therapy, as well as in the endometrium of healthy women and in the EC specimens obtained after hysterectomy (a comparison group). This group served as an additional control for the obtained results.

All genes were found to have increased methylation levels in EC. The analysis of the treatment outcomes for endometrial cancer IA stage G1 showed that the methylation level of *CDO1* and *CDH13* genes may help to predict a positive treatment outcome (complete response) with high accuracy (AUC = 0.96). We could not assess statistical differences in methylation levels between various subgroups with non-complete response, because we could not find a sufficient number of participants within the four-year study period; this is a certain limitation of our study. In the comparison group, the values of methylation levels of all studied genes were intermediate between the corresponding levels in EC CR and non-CR groups; this additionally confirms the validity of the obtained results.

The mechanisms by which the methylation of the *CDO1* and *CDH13* genes influence medical treatment outcomes are not yet clear; however, some literature data may provide insight into this issue.

The methylation of CpG sites of exon 1 is known to be tightly linked to transcriptional silencing. For other gene regions, this link is not so obvious [11]. In this study, we selected primers for CpG islands, overlapping exon 1, to analyze the methylation levels of *CDO1* and *CDH13* genes. Thus, it can be assumed that the increased methylation level found in our study would result in a reduction in gene expression.

Progestins can induce the apoptosis of EC cells; according to McGlorthan et al., progesterone induces apoptosis by the activation of caspase-8 in EC cells [12]. It can also be assumed that a slower proliferation of EC cells, exposed to progesterone [13], may contribute to their elimination by the immune cells. Inflammatory cancer is known to be the most aggressive form of cancer [14]. The association between inflammation and the activity of the disease may also apply to benign lesions. We have previously shown that the severity of external genital endometriosis inversely correlates with immune activity; however, in a small percentage of observations, the most aggressive forms of endometriosis were developed against a background of high proinflammatory activity [15]. In the absence of treatment, immune cells do not have enough time to eliminate actively proliferating EC cells. It is known that immune cells, particularly T cells, use the Fas ligand (FasL or CD95L) and TRAIL (TNF-related apoptosis-inducing ligand) to activate apoptosis in cancer cells [16]. A decreased expression of *CDH13* genes can lead to a reduced cell apoptosis, induced by various signaling pathways. For example, the *CDH13* promoter methylation was shown to increase the viability of non-small cell lung cancer cells exposed to cisplatin, by blocking DNA damage-induced apoptosis [17]. Studies of the effect of *CDO1* methylation on the survival of cancer cells have shown that the suppression of *CDO1* expression inhibits ferroptosis in gastric cancer cells. At the same time, this restores the level of cellular GSH and the content of malondialdehyde, one of the end products of lipid peroxidation, is reduced [18]. IFN-γ produced by CD8+ T cells is known to induce the ferroptosis of cancer cells by binding to IFNγR and activating several signaling pathways. These data indicate that a decreased CDO1 expression in EC cells may enhance their survival during their interaction with immune cells [19]. In AEH, only *CDO1* gene showed an increased methylation level. We were not able to assess the methylation levels in patients with AEH non-responsive to treatment, since in our study all patients had a complete response. This issue requires further research.

## 4. Materials and Methods

An observational cohort study of patients diagnosed with atypical endometrial hyperplasia (AEH) and endometrial cancer (EC) IA stage G1, subjected to conservative treatment, was conducted. Patients were followed from 2019 to 2023. The study included 41 patients: patients with AEH (*n* = 13) and patients with EC (*n* = 28). Inclusion criteria were as follows: no signs of invasion into the muscular layer according to contrast-enhanced magnetic resonance imaging (MRI) or ultrasound examination, and no signs of metastatic lesions in lymph nodes or ovaries. The patients willing to have a pregnancy were consulted by a reproductologist before therapy. In addition, each patient was informed that the recommended type of treatment was not standard for endometrial cancer and gave their consent. The patients diagnosed with AEH were treated with a levonorgestrel-releasing intrauterine device (LNG-IUD); the patients diagnosed with endometrial cancer IA stage G1 were treated with LNG-IUD and goserelin 3.6 mg per month. The initial treatment period was 6 months. According to the treatment outcomes, patients were divided into two groups: a group with a complete response (CR) and a group with a non-complete response (non-CR). The latter group was subdivided into four subgroups: a group with partial response to treatment, a group with stable disease, a group with the disease recurrence within a year after a successful 6-month course of treatment, and a group with disease progression under treatment. All patients who had a complete or partial response to the treatment and wished to conceive were offered IVF&ET. The study also included patients in peri- and early postmenopause with confirmed uterine cancer IA stage G1, who underwent radical treatment by hysterectomy with adnexa (comparison group, *n* = 18). The control group included endometrial samples from healthy women after diagnostic curettage for missed abortion and/or intrauterine adhesions (*n* = 18).

The response of AEH and EC to hormonal treatment was assessed pathomorphologically. The presence of the hormonal atrophy of all endometrial glands and a decidual reaction of the stroma, and/or the presence of typical glandular hyperplasia complexes was considered as complete response in both conditions. For endometrial cancer, a partial response was indicated by the presence of atypical endometrial complexes. EC was considered stable in the case of residual carcinoma complexes of the same degree. A decrease in the degree of cancer differentiation was considered as the sign of EC progression. 

The methylation study was carried out in the cytology laboratory of the Federal State Budgetary Institution “The Research Center for Obstetrics, Gynecology and Perinatology named after Academician V.I. Kulakov” of the Ministry of Health of the Russian Federation. The methylation level was assessed using a modified MS-HRM method described in the article by Krasnyi A.M. et al. [20]. The primers for analysis are presented in Table 3.

The amplification of fragments of CpG islands of the studied genes was performed according to the following protocol: 95 °C—5 min; (95 °C—15 s, 60 °C—30 s, 72 °C—45 s) × 30; (95 °C—15 s, 50 °C—30 s, 72 °C—45 s) × 25.

Statistical analysis was carried out in IBM SPSS Statistics 20. The significance of differences between the studied groups was assessed using the Mann–Whitney U test. The data are presented as median values, 1 and 3 quartiles (M (Q1; Q3)). We applied logistic regression and receiver operating characteristic (ROC) curve analysis, separately, to assess the diagnostic value of the studied parameters. The difference was significant at *p* < 0.05. The figures were plotted using OriginPro 8.5.

## 5. Conclusions

Thus, we have shown that an assessment of the *CDO1* and *CDH13* gene methylation levels in endometrial specimens from patients with endometrial cancer (IA stage G1) can predict the treatment outcome. The DNA methylation changes in AEH in the absence of a complete response to treatment could not be assessed and require additional study. Also, in the case of repeated treatment courses, it is important to evaluate dynamic methylation changes; this may allow a decision to be made on their feasibility for disease follow-up. In addition, our study indicates the potential value of genome-wide methylation assessment for selecting an optimal panel of prognostic markers for treatment outcomes.

## Figures and Tables

**Figure 1 ijms-25-04892-f001:**
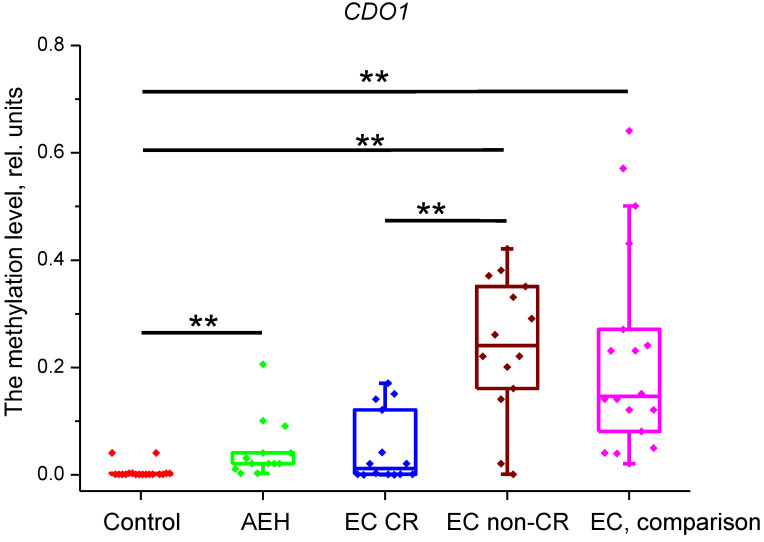
Methylation level of *CDO1* gene in the studied groups. **—*p* < 0.001. EC—endometrial cancer; AEH—atypical endometrial hyperplasia; CR—complete response.

**Figure 2 ijms-25-04892-f002:**
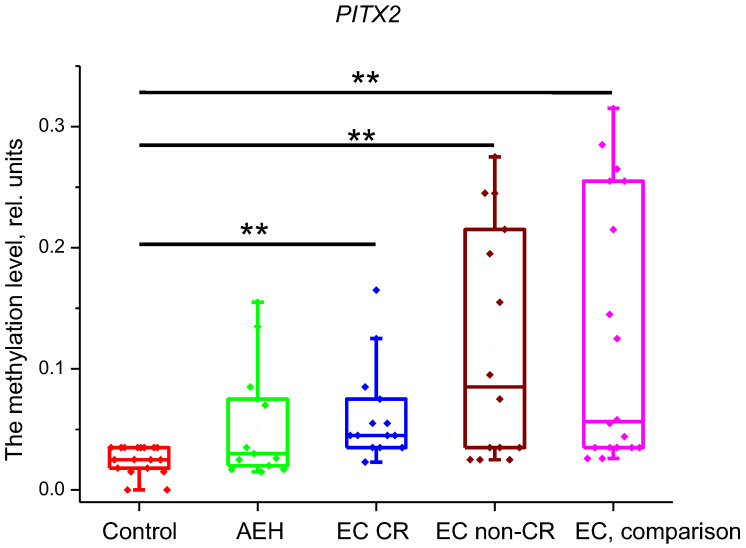
Methylation level of *PITX2* gene in the studied groups. **—*p* < 0.001. EC—endometrial cancer; AEH—atypical endometrial hyperplasia; CR—complete response.

**Figure 3 ijms-25-04892-f003:**
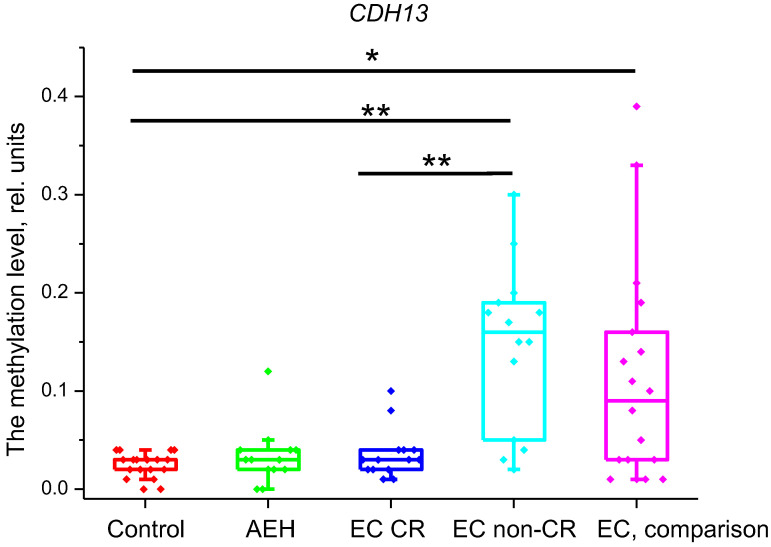
Methylation level of *CDH13* gene in the studied groups. *—*p* < 0.05, **—*p* < 0.001. EC—endometrial cancer; AEH—atypical endometrial hyperplasia; CR—complete response.

**Figure 4 ijms-25-04892-f004:**
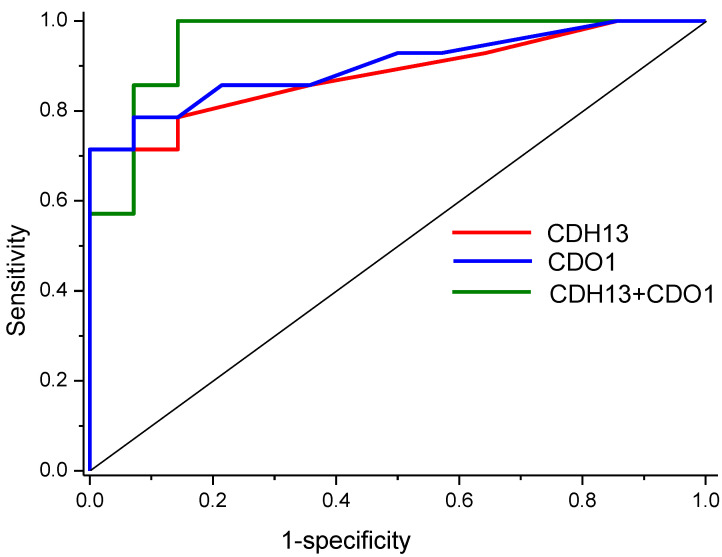
ROC curves for prognostic markers of EC treatment outcomes.

**Table 1 ijms-25-04892-t001:** Age and BMI of patients by groups.

Clinical Data	Control	AEH	EC CR	EC non-CR	EC Comparison Group	*p* *
Age	30 (33; 34.4)	37 (34; 42)	36 (34.5; 41)	34 (29.5; 34.5)	57.5 (42; 62.7)	0.01
BMI	32 (30.4; 32.9)	25 (24; 30)	31.5 (30; 32.5)	32 (30; 33.5)	25.5 (22.5; 33.7)	0.54

* *p*-value is presented for EC groups with complete response and non-complete response. EC—endometrial cancer; AEH—atypical endometrial hyperplasia; CR—complete response.

**Table 2 ijms-25-04892-t002:** Reproductive outcomes by disease.

Disease/Reproductive Outcome		Childbirth	Missed Abortion	Pregnancy Not Considered
AEH (*n* = 13)		3	2	8
EC CR (*n* = 14)		4	5	5
EC non-CR (*n* = 14)	No effect after 6 months of treatment; or disease progression; or disease recurrence with partial response to additional treatment courses (*n* = 10)	-	-	-
	Disease recurrence or partial response to the initial treatment course: additional treatment courses resulted in complete response (*n* = 4)	0	4	0

EC—endometrial cancer; AEH—atypical endometrial hyperplasia; CR—complete response.

**Table 3 ijms-25-04892-t003:** Primers for MS–HRM.

Gene	Amplicon Length	Number of CpG Sites	Forward Primer	Reverse Primer
*CDO1*	396	25	GGGAGGATGAATTTTATAGATTTG	TAAACTTCCATAATAACCTACACCT
*PITX2*	459	38	GTAGGAAGGAAATTAGAATTAAAT	AAAACTTACTACTAACTACCTCTTTTC
*CDH13*	495	33	GGGGTTTTTTTGTTTTTAGATT	CTTATCCACCCACTTACAAACTAC

## Data Availability

The data that support the findings of this study are available from the corresponding author, upon request.

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
