# Peer review of "Analysis of CDO1, PITX2, and CDH13 Gene Methylation in Early Endometrial Cancer for Prediction of Medical Treatment Outcomes"

_ijms, 2024, doi:10.3390/ijms25094892_

Round 1

Reviewer 1 Report

Comments and Suggestions for Authors

This article using patient samples to analyze the relationship between methylation of CDO, PITX2, and CDH13 and medical treatment outcomes. This study is of highly clinical significance and can guide the treatment of early endometrial cancer. However, there are some issues need to be improved.

1.     Authors should give convincing reasons of choosing CDO, PITX2, and CDH13 as predictive markers.

2.     For Tables 2 and 3, authors should provide more introduction and explanation, such as the impact of patient BMI on this study, the relationship between disease and reproductive outcomes and the study.

3.     Authors should follow abbreviation rules.

Comments on the Quality of English Language

English need to be improved.

Author Response

Dear Editor,

Thank you for your comments, they really helped to improve our article.

Following your recommendations, we have added the rationale for choosing CDO, PITX2, and CDH13 and the discussion for Tables 2 and 3.

Best regards,

Aleksey M., Krasnyi

Reviewer 2 Report

Comments and Suggestions for Authors

1.        Please explain the unit used in measuring the methylation levels.

2.        Please provide actual values of all the comparisons either in the text or summarize them in a table.

3.        Please add more in discussion. For example, please discuss more about the role of gene methylation in tumorigenesis, tumor progression, and treatment response. Is the scale of differences in methylation in this study a significant one based on previous studies? And please discuss the potential mechanisms as how CDO1 and CDH13 methylation could regulate treatment response in EMC.

Comments on the Quality of English Language

Minor editing needed. 

Author Response

Dear Editor,

Thank you for your comments, they really helped to improve our article.

Following your recommendations, we have added the rationale for choosing CDO, PITX2, and CDH13; expanded the discussion as advised, and added the actual values of all the comparisons in the text. The methodology is described in detail in the article we refer to, so we decided not to make any additions to it.  

Best regards,

Aleksey M., Krasnyi

Reviewer 3 Report

Comments and Suggestions for Authors

This manuscript essentially presents a proof-of-concept or small pilot study investigating CDO, PITX2 and CD13 gene methylation in early endometrial cancer. The article is clearly written and the figures are well-presented. Referencing is sparse, which is in keeping with the preliminary nature of the work and English language is generally good. The authors acknowledge that there are too few patient samples available to draw conclusions but it reasonable to propose that results obtained so far indicate that further studies could be pursued if additional samples were obtained. At this stage I feel that there is insufficient data to present as a full paper but it would be suitable to present as a short communication or letter, especially if it is unlikely that additional samples can be obtained by these authors. The authors could consider the following comments.

1. In the Introduction, the rationale for choosing to examine the CDO1, PITX2 and CDH13 genes for analysis in this study is not clearly stated, in particular in relation to endometrial cancer.

2. There are two small errors in Figure 1 (‘RC’ should be ‘CR’).

3. In the Discussion section (lines 148-149), the statement ‘probably, the methylation level would be higher in patients with subsequent negative treatment outcomes’ is not suitable in a scientific publication.

4. The half-page Discussion section in general is not presented as a discussion (which is likely a reflection of the small amount of preliminary data included in the manuscript). An alternative format journal where the Results and Discussion are combined may be more suited to this work.

Comments on the Quality of English Language

There are minor English language errors that will require correction.

Author Response

Dear Editor,

Thank you for your comments, they really helped to improve our article.

We have taken into account your recommendations: we added the rationale for choosing CDO, PITX2, and CDH13 and expanded the discussion. We have also made corrections as advised in Items 2 and 3 of your review.

Best regards,

Aleksey M., Krasnyi

Round 2

Reviewer 3 Report

Comments and Suggestions for Authors

The authors have made some amendments to the manuscript following review and this has improved aspects of the text. I still feel that the manuscript is disproportionately long in comparison to the small amount of data taken from too few patients to be statistically meaningful. Some speculative (non-scientific) and unreferenced statements remain, some of the Methods details are missing and the presentation of the tables could be improved. English language editing should also be applied.

1. With regards to selection of genes to include in the present study, the authors state that methylation of these genes was “mentioned in some studies” (lines 87-88). More precise details and the appropriate references should be added.

2. The authors have used a previously reported ‘in-house’ method to analyse gene methylation. In line 243 it is stated as reference 11, which is an error. The reference appears to be reference 16 in the current version of the manuscript. For the 3 genes that they are studying, is there evidence that the level and location of gene methylation that they are detecting is associated with loss of expression of these genes/encoded proteins? These references should be added to the text. The authors have also not stated how many CpG sites are included in the gene fragments amplified from each of the genes and whether they detected different levels of methylation of each of the gene fragments. More comprehensive details of the methods and results should be added to the manuscript.

3. In this study, the authors include an ‘EC, comparison’ group of women diagnosed with the same stage of endometrial cancer as the study group but who underwent radical (surgical) rather than conservative treatment. It is interesting that the methylation profiles of this group seem to mirror that of the endometrial cancer patients who did not achieve a complete response to conservative treatment (EC non-CR). How do the authors interpret these results?

4. Abbreviations used in tables should be explained as a footnote to each table.

5. All speculative statements require support either from results of this study or from previous publications. I have not listed all examples in this review (the authors need to do this), but statements such as “It can also be assumed that a slower proliferation of EC cells, exposed to progesterone, may contribute to their elimination by the immune cells. In the absence of treatment immune cells do not have enough time to eliminate actively proliferating EC cells” require back-up from previously published studies that have specifically examined this theory. (The example above is from lines 189-191).

6. All references need to be checked, including citation of the correct references in the text (see comment 2).

Comments on the Quality of English Language

English language editing will be required to improve grammar.

Author Response

Dear Reviewer,

Thank you very much for your very thorough re-review of the manuscript. You have raised very important questions which we tried our best to answer in the most attentive and comprehensive way.

Concerning the first comment:

A small number of participants is typical for most studies aimed at predicting outcomes of medical treatment of endometrial cancer. The most cited publication on this issue included only 9 patients. (Koji Yamazawa et al, reference #4 in the manuscript).

The answers to other comments:

  1. We studied the genes that are generally accepted as TSGs, as mentioned in many publications without reference to any primary source; for example, “p16 tumor suppressor gene” is found 2830 times in scholar.google search.
  2. In the Discussion section, we have added information about the correlation between methylation and gene expression. We also added the product size and the number of CpG sites for each gene. The methodology is described in detail in the referenced article, so we decided not to add redundant information to our manuscript.
  3. The EC comparison group included the patients in peri- and early postmenopause. The patients of advanced age routinely undergo only radical operations, since this option is most rational and safe for them, we assumed that the patients of this group could be either completely responsive (CR), or irresponsive to medical treatment (non-CR). This issue has already been discussed in the Discussion section.
  4. We have added the abbreviations’ explanations to the tables.
  5. For substantiation of our arguments, we have provided additional explanations and references in the Discussion.
  6. All references have been checked.

Round 3

Reviewer 3 Report

Comments and Suggestions for Authors

No comments 

Comments on the Quality of English Language

Minor editing only

Author Response

We have made changes based on comments from an academic editor.